# Evaluation of Quality, Antioxidant Capacity, and Digestibility of Chickpea (*Cicer arietinum* L. cv Blanoro) Stored under N_2_ and CO_2_ Atmospheres

**DOI:** 10.3390/molecules26092773

**Published:** 2021-05-08

**Authors:** Liliana Maribel Perez-Perez, José Ángel Huerta-Ocampo, Saúl Ruiz-Cruz, Francisco Javier Cinco-Moroyoqui, Francisco Javier Wong-Corral, Luisa Alondra Rascón-Valenzuela, Miguel Angel Robles-García, Ricardo Iván González-Vega, Ema Carina Rosas-Burgos, María Alba Guadalupe Corella-Madueño, Carmen Lizette Del-Toro-Sánchez

**Affiliations:** 1Department of Research and Postgraduate Studies in Food, University of Sonora, Rosales and Niños Heroes Avenue S/N, Hermosillo 83000, Sonora, Mexico; liliana.pz.1.9@gmail.com (L.M.P.-P.); saul.ruizcruz@unison.mx (S.R.-C.); javier.cinco@unison.mx (F.J.C.-M.); francisco.wong@unison.mx (F.J.W.-C.); qbc.ricardo.vega@gmail.com (R.I.G.-V.); carina.rosas@unison.mx (E.C.R.-B.); 2CONACYT-Research Center for Food and Development, Gustavo Enrique Astiazaran Rosas Road 46, Hermosillo 83304, Sonora, Mexico; jose.huerta@ciad.mx; 3Department of Chemical Biological Sciences, University of Sonora, Rosales and Niños Heroes Avenue S/N, Hermosillo 83000, Sonora, Mexico; luisa.rascon@unison.mx (L.A.R.-V.); mariaalba.corella@unison.mx (M.A.G.C.-M.); 4Cienega University Center, University of Guadalajara, University Avenue 1115, Ocotlan 47820, Jalisco, Mexico; miguel.robles@academicos.udg.mx

**Keywords:** chickpea, controlled atmospheres, antioxidant capacity, in vitro digestion, phenolic compounds

## Abstract

The aim of this work was to monitor the quality, antioxidant capacity and digestibility of chickpea exposed to different modified atmospheres. Chickpea quality (proximal analysis, color, texture, and water absorption) and the antioxidant capacity of free, conjugated, and bound phenol fractions obtained from raw and cooked chickpea, were determined. Cooked chickpea was exposed to N_2_ and CO_2_ atmospheres for 0, 25, and 50 days, and the antioxidant capacity was analyzed by DPPH (2,2′-diphenyl-1-picrylhydrazyl), ABTS (2,2′-azino-bis-[3ethylbenzothiazoline-6-sulfonic acid]), and total phenols. After in vitro digestion, the antioxidant capacity was measured by DPPH, ABTS, FRAP (ferric reducing antioxidant power), and AAPH (2,2′-Azobis [2-methylpropionamidine]). Additionally, quantification of total phenols, and UPLC-MS profile were determined. The results indicated that this grain contain high quality and high protein (18.38%). Bound phenolic compounds showed the highest amount (105.6 mg GAE/100 g) and the highest antioxidant capacity in all techniques. Cooked chickpeas maintained their quality and antioxidant capacity during 50 days of storage at 4 and −20 °C under a nitrogen atmosphere. Free and conjugated phenolic compounds could be hydrolyzed by digestive enzymes, increasing their bioaccessibility and their antioxidant capacity during each step of digestion. The majority compound in all samples was enterodiol, prevailing the flavonoid type in the rest of the identified compounds. Chickpea contains biological interest compounds with antioxidant potential suggesting that this legume can be exploited for various technologies.

## 1. Introduction

Mexico is one of the main chickpea producers in the world [1]. The chickpea’s quality for commercialization, such as its nutritional properties (carbohydrates, proteins, ashes, fats), color, and texture, influence storage and consumer acceptability. There are two types of chickpea, Desi, destined to be fodder, and Kabuli, destined for human consumption [2]. Consumption of Kabuli chickpea legume presents benefits to human health due to nutritional factors as a good source of proteins, minerals, vitamins, and bioactive compounds [3,4,5,6]. However, recently bioactive compounds such as phenolic compounds have taken an important role in foods due to the advantages to prevent degenerative diseases derived from their antioxidant capacity [7,8]. Unfortunately, these types of compounds are susceptible to degradation due to the presence of oxygen during storage. An alternative for this is the use of controlled atmospheres. It consists of a storage method in which the concentrations of oxygen, carbon dioxide, and nitrogen are regulated [9]. This technology is a more natural, low-cost alternative, convenient to use and environmentally friendly [10]. The efficacy of the controlled atmospheres in foods has been elucidated in several studies [10,11,12,13]. However, few studies are carried out in legumes, especially in cooked chickpea.

Additionally, phenolic compounds can be in the food matrix in three different forms: (a) The free phenolic compounds linked superficially by hydrophobic interactions, hydrogen bridges, or electrostatic forces [14]; (b) Conjugated phenolic compounds linked to proteins and sugars by weak bonds similar to free phenols [15]; (c) Bound phenolic compounds covalently linked by ester or ether bonds to the cell wall with components such as cellulose, hemicellulose, lignin, pectin, and structural proteins [15]. However, the determination of total phenolic compounds in legumes in most of the studies is underestimated because, in the extraction process for their quantification, the free phenolic compounds are generally extracted and sometimes the conjugates, while the bound phenolic compounds are not considered [4,7]. Thus, it is important to consider them because the bound phenolic compounds can be found in greater quantities (approx. 70%) and consequently could confer greater antioxidant activity [16,17]. However, this can vary depending on the type or variety of chickpea [11]. 

To take advantage of chickpea antioxidant capacity benefits, phenolic compounds must have high bioaccessibility (the fraction potentially available for further uptake and absorption) in the small intestine [18]. In vitro digestibility simulation technique can help to determine the phenolic compounds transit during each step of the digestibility (mouth, stomach, and small intestine) and if the phenolic compounds can release in each step. The multiple advantages of in vitro research, such as being fast and cheap, better condition control, and less ethical restrictions, have made it an alternative method to studying the metabolic process of biologically active substances in vivo [19]. Recently, studies have focused on determining the bioaccessibility and bioavailability of phenolic compounds in legumes through static or dynamic in vitro models. The static model preferably is used if the focus of research is on hypothesis building and a large number of samples is to be analyzed, while the dynamic model is used if closely simulating physiological conditions is the primary aim [20]. For polyphenols, there is limited evidence as to which method is the most appropriate for measuring bioaccessibility. However, to test the bioactivity of these antioxidant compounds after in vitro digestion some antioxidant techniques such as DPPH (2,2′-diphenyl-1-picrylhydrazyl), ABTS (2,2′-azino-bis-[3ethylbenzothiazoline-6-sulfonic acid]) and FRAP (ferric reducing antioxidant power) have been used [19,21,22]. The determination of phenolic compounds in each step of digestion is commonly used to measure their bioaccessibility [23,24]. Bioavailability has been studied by quantifying these compounds after being absorbed by a cellulose membrane used to simulate the small intestine [25,26] or also using Caco-2 cell monolayers [21,27].

On the other hand, alpha-amylase (mouth), pepsin (stomach), chymotrypsin, trypsin, and pancreatin (small intestine), are enzymes that have been used in the in vitro digestion system in many studies [23,24,28,29,30]. These enzymes can hydrolyze covalent bonds from sugar and proteins. Consequently, this could confer greater bioaccessibility to the different types of phenolic compounds. Therefore, this study aimed to determine the quality and antioxidant capacity of free, conjugated and bound cooked chickpea Kabuli-type under atmospheres of N_2_ and CO_2_ and its digestibility through a gastrointestinal system in vitro.

## 2. Results and Discussions

### 2.1. Proximal Analysis

The differences in composition between raw and cooked chickpea are important because they provide insight into the components that are modified or changed after cooking. Table 1 shows the proximal composition of these samples. In general, carbohydrates are the majority constituent group of legumes-grain. The carbohydrate content showed no significant differences between raw (64.40%) and cooked (68.32%) chickpeas. Those values are in accordance with those found in other reports, and that are in a range of 51 to 75% [4,5,6,31,32]. Moisture, ash, protein, and fat presented significant differences (*p >* 0.05) between raw and cooked samples. 

The moisture was higher in the cooked (47.28%) than the raw chickpea (5.96%) because the chickpea got water in the process when was soaked. The raw chickpea values are in the range of those reported in the references (5.48–10.7%) [5]. In cooked samples, some studies reported 56–100% [10,11], where these values are higher than our samples. Probably, the moisture content can vary considerably due to the soaking time, the cooking and even the variety of the chickpea, ground floor type, way of cultivation, and management conditions for its development [3,33,34].

On the other hand, the decrease in ash, fat and protein content could be affected by the soaking or temperature. The ash content is a measure of the total minerals; probably, minerals are lost by leaching in the soaking and cooking waters [35]. Similar results were observed in the study of Marconi et al. [36], where reduction in ash was observed in beans and chickpeas after cooking. In general, the ash value for raw chickpeas is in a range of 2.5 to 3.4% [4,5,6], the ash content obtained in this study are within this range (2.8%). According to Table 1, the fat of chickpea cooked (5.35%) was lower than the raw sample (6.23%). The amount of legume fat may be important in the formation of the amylose-lipid complex and have good performance in the gelatinization of starch during cooking, a phenomenon that tends to limit the bioavailability of starch [13]. Comparing with that obtained by other authors in different cultivars (4.64–7.24%) [4,5,6], the fat content of raw chickpea (5.10%) in our study is in this range. Regarding the protein content, the reports on the amount of protein in chickpea are from 7.8 to 51.94 [3,4,5,6]. In the present study, the raw chickpea protein amount (19.16%) was high. This protein decreased after cooking (18.55%), probably due to a solubilization effect of seed proteins in water [37]. However, the amount of protein left in cooked chickpeas is still considered high compared to other raw legumes with the same values.

Finally, the fiber refers to the quantification of fibrous elements of the wall of the plant cell. The amount of raw chickpea fiber was 7.41%, and there was no significant difference in its content after cooking. This amount is in the range reported by other authors (5.36–12.20%) [5,6] in Kabuli-type chickpea.

### 2.2. Technological Quality

The process of hydration (soaking) of legumes is important because it has several functions: (a) Reduces cooking time; (b) Allows for the initiation of enzymatic activity that makes them more digestive and (c) It is necessary to soften them and then they can be cooked and eaten easily. The water absorption and texture of chickpea are shown in Figure 1. It is shown that the sorption behavior is of the logarithmic type with increasing water content during soaking time. This increase was until 8 h, probably due to filling free capillaries and inter-micellar spaces on the seed coat and hilum [38]. From 8 to 22 h of soaking, the water absorption did not show significant differences (*p* > 0.05). According to other research, the saturation point of water absorption in chickpea has been reported between 4 to 24 h [10,39,40]. Regarding the texture, the grain’s hardness was decreasing (from 0.69 N to 0.28 N), with no significant differences after 6 hours of soaking. Similar behavior was observed by Kumar et al. [11] where they reported in two different varieties of raw chickpea values of 0.15 N and 0.16 N. Therefore, the determination of water absorption and the hardness (texture) during soaking was an indicator of the time (8 h) when the grain absorbs the most water with less hardness, being able to reduce the cooking time.

Among the various characteristics of high-quality chickpea, color is one of the most important since it influences the selection of new, improved varieties of chickpea in experimental research centers and also the price on the international market [39]. There is currently no classification of the colors representing chickpeas’ grain as an objective reference when marketing. It has been so far the perception of the consumer. In Table 2 is shown the results of color between the chickpea raw and cooked. The parameter lightness (L*) was higher in the raw chickpea due to the oxidation which causes a darkening of the sample and, consequently, the lightness in the cooked sample. In the chromaticity diagram, the value of a* chickpea cooked was higher than the raw chickpea, which means that the value has tones closer to the red tones than the raw sample; therefore, the cooked chickpea is more intense than the raw. The same happens with the parameter b*, where the cooked chickpea has a higher value, which corresponds to a yellowish tone, than the raw chickpea. Similar behavior was reported by Reyes et al. [41]. However, chickpea colors can vary considerably. Some studies have reported differences in the color between the Desi and the Kabuli chickpea, and even within its own varieties [11,42] 

### 2.3. Modified Atmospheres Test

The modified atmospheres can help preserve different compounds, especially those with bioactive importance from food, such as phenolic compounds that possess the antioxidant capacity. The antioxidant capacity determined with DPPH (Figure 2) and ABTS (Figure 3), as well as total phenolic compounds (Figure 4) from cooked chickpea under air (control), N_2,_ and CO_2_ atmospheres, were monitored for 0, 25 and 50 days at different temperatures in each phenolic fraction (free, conjugated and bound). In general, it can be observed that all the samples under nitrogen atmospheres are better preserved than in the rest of the atmospheres used. The antioxidant capacity and the phenolic compounds were more stable at 4 and −20 °C during the storage time under the nitrogen atmosphere, with no significant differences (*p* > 0.05).

Initially, the antioxidant capacity determined by DPPH (Figure 2) of free, conjugated, and bound phenolic compounds were 19.76, 35.54, and 56.52% of inhibition, respectively. In the same samples, in ABTS (Figure 3), they started with 19.15, 26.3, and 92.17% of inhibition, respectively. These differences could be due to the phenolic content characterized by their antioxidant properties. The amount of free, conjugated, and bound phenolic compounds (Figure 4) were 22.04, 19.43, and 105.6 mg GAE/100 g of the sample, respectively. Bond phenols are the highest in quantity, possibly because they are more protected within the food matrix as they are attached to polysaccharides such as pectin, cellulose, and hemicellulose. That could explain the higher antioxidant potential of bound phenols in both radicals. However, each radical can present different affinities for the sample and consequently give different values despite being the same sample. In this context, ABTS reacts more with hydro- and lipophilic compounds and an H-atom donor [43,44]. While DPPH has less affinity for compounds containing aromatic rings with only hydroxyl groups [45,46] and more affinity for lipophilic compounds [47].

The degradation of the antioxidant capacity and phenolic compounds is most evident in air and CO_2_ atmospheres from day 25 to 50, showing a maximum degradation of approximately more than 87.84% under these conditions. While in the N_2_ atmosphere, the maximum degradation was approximately 32.8% at day 50. That could be because the N_2_ replaces the O_2_ of the container’s interior, avoiding oxidative degradation in the food to maintain the high levels of antioxidant capacity and phenolic compounds in the chickpea longer time [12,13]. Hence, the N_2_ a gas being totally inert and having extraordinarily low solubility in water and fats makes it an ideal product for preservation, preventing oxidation and future contamination in foods and beverages. Therefore, the atmosphere of N_2_ is the most suitable to preserve both antioxidant capacity and phenolic compounds of cooked chickpea during storage.

On the other hand, according to the results obtained during the chickpea storage, the samples in an atmosphere of N_2_ at −20 °C almost entirely retain the phenolic compounds and their antioxidant capacity. However, a proximal analysis of these conditions was performed to determine whether there were changes within the nutrient content until 50 days of storage (Table 3). It can be observed that there were no significant differences (*p* > 0.05) between the initial quantities and up to 50 days in carbohydrates, ash, protein, fat, and fiber. Similar results were found by Kumar et al. [11]. Hence, this chickpea can be considered a food with good nutritional quality because it is high in protein, fiber, and fat, with a total amount of mineral acceptable for legumes [48].

### 2.4. In Vitro Gastrointestinal Digestion

The results of the chickpea digestion are shown in Table 4. Firstly, before the in vitro digestion process, we compared the antioxidant capacity and the amount of free, conjugated, and bound phenolic compounds from raw and cooked chickpea. A decrease in the amounts of phenolic compounds (free 58.8%, conjugated 28.7%, and bound 26.9%) was observed after cooking. According with De Santiago et al. [30] and Mtolo et al. [24], a longer heat treatment (more than 15 min) applied to the food has structural consequences such as changes in the food matrix, and a decrease in phenolic compounds can occur. This same behavior was presented in the antioxidant capacity (DPPH: 23.6–51.2%, ABTS: 44.2–52.5%, FRAP: 33.8–59.5, and AAPH: 44.0–53.9%). This tendency occurs when phenolic compounds are responsible for conferring the antioxidant property to the sample. To verify this, Pearson’s correlations were determined (Appendix A). The Pearson’s correlation coefficients between the phenolic compounds (conjugated and bound principally) are strongly (r > 0.700) observed with the antioxidant capacity in all techniques used, except for the free phenolic fraction, which showed moderate correlation (r = 0.449) with FRAP in the raw chickpea. FRAP is based on the reducing power of an antioxidant (in this case, chickpea phenolic compounds) from ferric ion (Fe^3+^) to the ferrous ion (Fe^2+^). This electron transfer is directly proportional to the antioxidant activity. Thus, the moderate correlation indicates that free phenolic compounds do not fully confer this electron transport in raw chickpea. In this context, the Folin–Ciocalteu technique used for the determination of polyphenols can react with other compounds different from phenolics such as proteins, sugars and others with reducing power, which may be present in chickpea extracts [49].

It is important to mention that the DPPH and ABTS are synthetic radicals. However, they give an idea of chickpea phenolic compounds’ possible behavior with the radicals in the organism. On the other hand, AAPH is a generator of free radicals inducing oxidation of lipid membrane in the erythrocyte. This technique directly measures chickpea phenolic compounds’ protective effect against erythrocytes’ oxidative damage [50]. Under this context, bound phenolic compounds (Table 4) showed the most oxidative protection in raw (76.8% of inhibition) and cooked chickpea (43.0% of inhibition) with this assay. These bound phenolic compounds presented the most antioxidant capacity (raw and cooked chickpea) in all antioxidant techniques used. However, during the digestion process, the behavior of these compounds is different. Probably because the bound phenolic compounds are strongly bonded with cellulose, hemicellulose, and pectin of the chickpea matrix [51], the bioaccessibility of these compounds is very low from the mouth (alpha-amylase) to the small intestine (pancreatin). Therefore, these compounds must pass to the large intestine, where they will be metabolized by the intestinal microbiota producing lower molecular weight phenols with higher absorption. Consequently, these compounds’ antioxidant capacity will increase because the functional groups of phenolic compounds will be more exposed [2].

On the other hand, free and conjugated phenolic compounds could be hydrolyzed by digestive enzymes, increasing their bioaccessibility and consequently their antioxidant capacity during each step of the digestion. As these compounds are more bioaccessible, they have a better chance of being absorbed through the intestine (inside the membrane) and passing to the portal vein (outside the membrane) to be transported to the body’s different cells. For this reason, we can observe that when consuming cooked chickpea, around 30% of the free phenolic compounds and 45% of the conjugated compounds can be absorbed through the intestine conferring good antioxidant capacity (Table 4). It should be considered that in this in vitro study, there is no transport through the membrane or a microbiota, so these percentages could probably increase with normal digestion.

Some studies showed similar behavior related to the antioxidant capacity and phenols amount than our study [23,29]. However, other studies reported in FRAP no significant difference between the initial sample to the in vitro digestion sample [24]. Referring to the evaluation of the protective effect on human erythrocytes against AAPH, there are few studies about this topic in legumes. Zhao et al. [52] reported with this analysis free (3.46–44.27% of inhibition) and bound (0.29–8.56% of inhibition) phenolic fractions of different cereals, contrary to our results since we obtained bound phenolic compounds in a higher proportion with the highest antioxidant capacity.

### 2.5. Identification of Phenolic Compounds by UPLC-MS

Free, conjugated and bound phenolic compounds were identified by UPLC-MS in raw and cooked chickpea (Table 5 and Appendix A) and were compared with the Database on polyphenol content in foods [53]. Around twelve compounds were identified. The majority compound of free, conjugated, and bound phenolic compounds in both raw and cooked chickpea was enterodiol (46.4–67.42%), a lignan compound with several beneficial properties as an antioxidant and as an inhibitor of some types of cancer [54,55,56,57]. The rest of the compounds are found in varying amounts and belong to different classes of phenolics (Table 5 and Figure 5): (a) Lignans: Gomisin D (5.81–10.33%) and Anhydro-secoisolariciresinol (12.43–12.78%); (b) Flavonoids: Pelargonidin 3,5-*O*-diglucoside (1.95–8.67%), Hesperetin 3′,7-*O*-diglucuronide (5%), 6-Geranylnaringenin (7.46–7.73%), Isorhamnetin (1.55%) and Phloretin 2′-*O*-glucoside (1.67%); (c) Phenolic acids: *p*-Coumaroyl glucose (2.90%) and 3,4-Diferuloylquinic acid (6.61%); (d) Other polyphenols: Hydroxytyrosol 4-*O*-glucoside (3.08–5.62%) and 1,4-Naphtoquinone (3.54%). The presence of more variety of phenolic compounds of the flavonoid type can be observed. That is characteristic of grains and legumes’ phenolic composition [58,59].

On the other hand, there are phenolic compounds in glycosidic form, especially in free phenolic compounds. Von Gadow et al. [45] indicated that the antioxidant efficiency is directly related to the degree of hydroxylation, but this decreases with the presence of a sugar bond. That may explain the low antioxidant capacity observed in the free phenolic compounds compared to the bound and conjugated compounds. Additionally, it has been found that phenolic compounds with the highest number of substituents, mainly hydroxyl and those with the lowest number of methoxy substituents are more easily degraded by the effect of temperature [60]. That explains the decrease in phenolic compounds and their antioxidant capacity in cooked chickpea compared to raw.

The antioxidant capacity is not related proportionally to the individual quantity of phenolic compounds. One compound may exist in a smaller proportion than another and confer the highest antioxidant capacity of the sample. One characteristic that gives the antioxidant capacity is the hydroxyl groups, which are attached to a benzene ring, as in the case of polyphenols, present the possibility that the free radical interacts with ring electrons, which gives them special characteristics with respect to other alcohols. Therefore, it will depend on its structure, where it has been found that the greater the hydroxylation, the higher the antioxidant capacity [2,61]. Besides the hydroxyl groups, other factors make a molecular antioxidant, such as double bonds in the structure, hindrance steric, and the hydroxyl group’s position. Another factor is the affinity to the compounds to the radical or the technique.

On the other hand, most of the compounds identified after in vitro digestion (Table 6 and Appendix A) are different from those observed before the digestion process. Probably, the enzymes present in each part of the process contributed to the structural modifications. In the mouth section, compounds without sugars were observed, such as caffeoyl tartaric acid, *p*-coumaroyl tyrosine, syringaldehyde, and glycitein (Figure 6). That means that the enzymes as beta-glucosidase and alpha-amylase from saliva can liberate these sugars from phenols [62]. Surprisingly, it was not possible to identify compounds released in the mouth in the conjugated phenolic compounds, perhaps because the food is not long in the mouth and probably the interaction with the enzymes was not sufficient to react with these compounds. In the stomach, the lower pH can induce phenolic compounds hydrolysis, especially the phenolics that are joined to polysaccharides that have, consequently, liberated the phenolic compounds, as is the case of conjugated phenolic compounds (Table 6). However, in free and bound phenolic compounds, some sugar aggregates were observed (Figure 6), maybe because the phenolic compounds can destabilize the protein structure, reducing the accessibility of the enzyme, decreasing hydrolysis, and allowing free sugars to rejoin phenolic structures.

In addition to the compounds found in the small intestine fraction (Table 6 and Figure 6), they are compounds with high antioxidant capacity (flavanols: [(-)-Epicatechin-(2a-7) (4a-8)-epicatechin 3-*O*-galactoside]; tyrosols: [tyrosol 4-sulfate]; anthocyanins: [Pelargonidin 3-*O*-glucosyl-rutinoside and Delphinidin 3-*O*-(6″-acetyl-glucoside)]; isoflavonoids: [Pseudobaptigenin] and hydroxycinnamic acids: [Caffeoyl glucose]). In this part, phenolic compounds binding can hinder substrate binding site, catalytic site, or both of the pancreatin, thereby reducing its proteolytic activity and letting the free amino and thiol groups and tryptophan residues, which came to the covalent attachment of the phenolic compounds [23,63]. This enzyme can decrease its activity due to the position and hydroxyl groups of the phenolic compounds, but with all those changes can form different quinones that are more reactive. The phenolic compounds that could not pass through the membrane of the small intestine may be due to these phenolic compounds not released from the matrix and going directly to the large intestine, as is the case with most bound phenolic compounds. The reason is that they are linked covalently to the matrix, mainly by ester bonds that are hydrolyzed by enzymes from the microbiota as esterases [2,64,65].

## 3. Materials and Methods

### 3.1. Chemicals and Reagents

Reagents such as gallic acid (3,4,5-trihydroxybenzoic acid), 2 N Folin–Ciocalteu phenol reagent solution, DPPH (2,2-diphenyl-1-picrylhydrazyl), ABTS (2,20-azino-bis-[3ethylbenzothiazoline-6-sulfonic acid]), AAPH (2,′2.Azobis [2.methylpropionamidine]), pepsin (P7012), pancreatin (P1750), TPTZ (2,4,6-tripyridyl-s-triazine), phosphate buffered saline (Sigma P-3813, Darmstadt, Germany), dialysis tubing (D-9527), chromatography standards, and HPLC-solvents were from Sigma-Aldrich Co. (Darmstadt, Germany). The rest of the chemical reagents were of analytical grade. Propylene bags were obtained from GrainPro (Concord, MA, USA).

### 3.2. Plant Material

Chickpea (*C. arietinum* var. Blanoro), Kabuli-type were provided in 2019 from Campo Covadonga in Hermosillo, Sonora, Mexico. The coordinates were longitude: −111.483056 W and latitude: 28.880556 N at 70 meters above sea level.

### 3.3. Preparation of Sample

The chickpea grains were cooked by the method of [66], with some modifications. The grains (500 g) were soaked in 2.5 L of water for 22 h and, after that, the grains were cooked by steaming for 60 min. During soaking, the water absorption and texture were determined as described below. The raw and cooked grains were analyzed in their proximal analysis and color. The controlled atmospheres study and digestibility were determined only in cooked samples. For the phenolic extractions (free, conjugated, and bound) and the antioxidant capacity analysis, the raw and cooked samples were lyophilized (Freeze Zone 4.5, Labconco, Kansas city, MO, USA) and milled in a Perten laboratory mill model LM3100 (PerkinElmer, Waltham, MA, USA) to a final particle size of 0.5 mm mesh.

### 3.4. Proximal Analysis

Raw and cooked chickpea were analyzed for crude protein (method 955.04) and ash (method 920.153) using the AOAC [67] methods. Total fiber (method 985.29), fat (method 920.85), and moisture (method 952.08) were determined according to AOAC [68]. Carbohydrates were determined by difference.

### 3.5. Technological Quality

#### 3.5.1. Water Absorption

Water absorption was evaluated before cooking, measuring seed weight during the soaking process. The absorption curve increased with the weight of the seeds as a function of time. A total of 15 seeds were taken by triplicate with 65 mL of water, and the measurements were carried out at 2 h intervals until 22 h. The percent of absorption was measured according to the following Equation (1) [69]:(1)$$\%\;\mathrm{water}\;\mathrm{absorption}=\frac{M_{t}-M_{0}}{M_{0}}\times 100$$
where M_0_ and M_t_. are the mass of chickpeas before soaking and at different soaking intervals, respectively.

#### 3.5.2. Texture

The texture was measured based on the methodology of Ponce et al. [70] by a Texturometer Instron 4465 (Instron Corporation, Canto, MA, USA). The contact area of an individual seed was measured seed by seed. The strain of the seed was 50% deformation with a speed of 0.2 mm/s. Results were expressed as Newtons.

#### 3.5.3. Color

The color was determined with a HunterLab Miniscan XE colorimeter Model (Hunter Association Laboratories, Reston, VA, USA). The seeds were homogenized using 17 lectures. The result was reported by luminosity (L*) evaluated on a scale of 100 (white) to 0 (black), *a** where +a* is red and −a* is green, and *b** indicate yellow (+b*) and blue (−b*) shades [71].

### 3.6. Controlled Atmosphere Conditions

Cooked chickpea grains (50 g) were placed in propylene bags GrainPro (11 × 15 cm) and were exposed to CO_2_ (100%), N_2_ (100%) and air (CO_2:_ 0.0%, O_2:_ 10.5%, N_2:_ 89.5%) as control through a gas mixer (Thermco, La Porte, IN, USA). The air in the bags was displaced with the gases; the bags were sealed until they reached the desired gases concentration. Gases were monitored using a gas analyzer (Viasensor, www.viasensor.com). The temperatures of the treatments were −20 °C, 4 °C, 25 °C, and 50 °C. The antioxidant capacity by DDPH, ABTS and total phenols (free, conjugated, and bound phenols) were monitored at 0, 25, and 50 days.

### 3.7. In Vitro Digestibility Test

The in vitro digestibility test simulates the mouth, stomach, and small intestine conditions to evaluate the phenolic compounds’ in vitro bioaccessibility from chickpeas and their antioxidant properties. The test was carried out according to Gil-Izquierdo et al. [72] with modifications. A total of 15 g of cooked chickpea was given to a healthy volunteer (who brushed his teeth with toothpaste, and the last bite he tasted was 90 min before the test) and chewed it 15 times for 15 seconds. The sample was homogenized in 10 mL of distilled water and acidified to pH 2 with 6 M HCl. A volume of 22.5 mL of pepsin (315 U/mL prepared in 0.2 M KCl buffer) and 22.5 mL of deionized water were added to the sample and shaken for 2 h at 80 rpm in a water bath (Wise Bath, DAIHAN Scientific, WSB-18) set at 37 °C. The mixture was neutralized with NaHCO_3_ 1.25 M, and 5.625 mL of pancreatin (4 mg/mL prepared in 0.1 M phosphate buffered saline) was added. After homogenization, the sample was placed inside a dialysis membrane (Sigma D9888-100FT; 12,000 Dalton cut-off) pre-conditioned (30 cm of the membrane were cut and left in distilled water for 1 day). The membrane was placed inside a test tube containing approximately 160 mL of phosphate buffered saline 1 M pH 7.4 and incubated again in the shaking water bath (4 h, 37 °C, 80 rpm). Antioxidant activity by DPPH, ABTS, FRAP, AAPH, and total phenols were determined before (initial) and after digestion (inside and outside of the membrane) from the extracts of free, conjugated, and bound phenolic compounds of the sample.

### 3.8. Extraction of Free, Conjugated and Bound Phenolic Compounds

Free phenols were extracted according to the procedure reported by Cabrera-Soto [73] with modifications. Raw and cooked samples (15 g) were homogenized in a stirring plate with 30 mL of water:methanol (20:80, *v*/*v*) for 1 h at 100 rpm. Subsequently, the sample was centrifuged at 4000 rpm for 15 min at 4 °C. The supernatant was collected, and the residue was used for a second extraction employing the same procedure. The supernatants were combined and evaporated in a rotary evaporator (RE301, Yamato, Santa Clara, CA, USA) at 45–50 °C until 15 mL. Then, 5 mL of distilled water was added, and it was washed twice with ethyl acetate at a 1:1 *v*/*v* ratio with the sample. The ethyl acetate phase was evaporated and reconstituted to a final volume of 2 mL with methanol (free phenolic extract). The water phase was kept for the conjugated phenols and the sediment for the extraction of bound phenols.

For the soluble conjugated phenols, the water phase was filled to 20 mL with distilled water. The application of high-energy ultrasound (Digital Sonifier 250; Branson, Mexico) has been used as an alternative to improve the phenolic extraction. The sample was ultrasonicated at 40% amplitude during 59 s with a time free of 30 s until complete 1 h at 20 kHz. During the sonication, samples were maintained in an ice bath. After two extractions with ethyl acetate (1:1 *v*/*v* solvent/sample), the sample was evaporated and reconstituted to a final volume of 2 mL with methanol (conjugated phenolic extract).

The sediment obtained in the free phenolic procedure was adjusted to 30 mL of water:methanol (20:80, *v*/*v*) to obtain the bound phenolic extract. The sample was ultrasonicated using the same conditions as the conjugated phenols. Subsequently, the sample was centrifuged at 4000 rpm for 15 min at 4 °C. The supernatant was collected, and the residue was used for a second extraction employing the same procedure. The supernatants were combined, and after two extractions with ethyl acetate (1:1 *v*/*v* solvent/sample), they were evaporated and reconstituted to a final volume of 2 mL with methanol (bound phenolic extract).

### 3.9. Determination of Antioxidant Capacity

The antioxidant test was measured with radical DPPH [74], ABTS [75], FRAP [76], AAPH [77], and total phenolic content [78].

#### 3.9.1. DPPH

Two hundred microliters of the methanolic solution of the radical DPPH (6 × 10^−5^ mol/L) were mixed with 20 µL of the sample. This preparation was measured at 515 nm in a microplate reader (Thermo Fisher Scientific Inc. Multiskan GO, Waltham, MA, USA) after an incubation time of 120 min in darkness. DPPH with the solvent was taken as control. The determinations were in triplicate. The results were expressed as nanomoles of Trolox equivalent per g of sample (nmol TE/g of sample) for the in vitro digestion analysis and as a percentage of radical inhibition for the controlled atmosphere study according to the following Equation (2):(2)$$\%\;\mathrm{of}\;\mathrm{inhibition}=\frac{\mathrm{inicial}\;\mathrm{absorbance}-\mathrm{final}\;\mathrm{absorbance}}{\mathrm{inicial}\;\mathrm{absorbance}}\times 100$$

#### 3.9.2. ABTS

The radical (ABTS^∙+^) was formed by dissolving 19.3 mg of ABTS in water (5 mL). An aliquot of 88 µL from a potassium persulphate solution (0.0378 g/mL) was added to the ABTS solution. The mixture was left for 12 h at room temperature in the dark. The ABTS^∙+^ solution was diluted with ethanol to an absorbance of 0.700 ± 0.050 at 734 nm. The sample (20 µL) was mixed with 270 µL of radical solution and was incubated in darkness for 120 min. Control and determinations at 734 nm were the same as in the DPPH assay.

#### 3.9.3. FRAP

Ferric reducing antioxidant power (FRAP) assay follows the reaction of Fe^3+^-TPTZ at 638 nm. First, the stock solutions were prepared: sodium acetate buffer (300 mmol/L, pH 3.6), FeCl_3_ (20 mmol) and TPTZ solution (10 mM) in HCl (40 mM). The FRAP working solution was prepared 10:1:1 (buffer: FeCl_3_: TPTZ). Then, a volume of 20 µL of the sample was combined with 280 µL FRAP solution and placed in a microplate reader (Thermo Fisher Scientific Inc. Multiskan GO, Waltham, MA, USA). After 30 min, the absorbance was read at 638 nm. FRAP working with the solvent of the sample was taken as control. The results were expressed as nanomoles of equivalent Trolox per g of sample.

#### 3.9.4. AAPH

In this assay, AAPH was used to induce free radicals’ formation from the erythrocyte membrane causing hemolysis. Therefore, it was intended to observe the sample’s antioxidant capacity by inhibiting the radicals formed by AAPH, thus avoiding hemolysis. First, erythrocytes were collected from a healthy blood type A+ adult volunteer of 30 years old using the Mexican (NOM-253-SSA1-2012) and international (FDA: CFR - Code of Federal Regulations Title 21, part 640) regulations. A technical committee accredited the laboratory where the sample was treated. The complete information about the regulations used was described by González-Vega et al. [50].

The globular package was separated from the plasma by centrifugation at 1500 rpm for 10 min at 4 °C. The erythrocytes were washed three times with phosphate-buffered saline (PBS) pH 7.4. Afterward, a suspension of erythrocytes was prepared in PBS with a ratio of 5:95 (*v*/*v*). A mixture containing 100 μL of the erythrocytes, 100 μL of the AAPH radical, and 100 μL of the sample was incubated at 37 °C for 3 h with continuous shaking (30 rpm). Subsequently, 1 mL of PBS was added and centrifuged at 1500 rpm for 10 min at 4 °C. A suspension of erythrocytes, and AAPH with erythrocytes, were taken as controls. The supernatant was read at 540 nm on a 96-well microplate. The percentage of inhibition was determined from the following Equation (3):(3)$$\%\;of\;inhibition=\left(\frac{\mathrm{AAPH}1-\mathrm{HS}}{\mathrm{AAPH}1}\right)\times 100$$
where AAPH_1_ = absorbance of hemolysis induced by AAPH; HS = absorbance of the sample.

#### 3.9.5. Total Phenolic Content

This spectrophotometric method was assayed using the Folin–Ciocalteu reagent in a 96-well microplate format. Briefly, 10 µL of the sample was mixed with 25 µL of 1 N Folin–Ciocalteu reagent. The mixture was incubated for 5 min at room temperature. Afterward, 25 µL of Na_2_CO_3_ (20%) and 140 µL of deionized water were added, and the absorbance was recorded at 760 nm. The reagent with the solvent of the sample was taken as control. A gallic acid standard calibration curve (0–100 mg/L) was prepared. The results were expressed as mg of gallic acid equivalents (GAE) per 100 g of sample.

### 3.10. Identification of Phenolic Compounds by UPLC-MS

The procedure was based on Hernández-Cruz et al. [79] with modifications. It was used Ultra Performance Liquid Chromatography (Waters Acquity 2690) (Waters Corp, Milford, MA, USA) with UV detector linked to mass spectrometry SQ Detector 2 simple quadrupole (Waters Corp, Milford, MA, USA) managed by Acquity console software with diode array detector and an automatic injector. The phenolic extracts were filtered using 0.22 µL filters and 20 µL were injected. The column was Acquity UPLC BEH C18 2.1 × 50.0 mm, 1.7 µm particle size. The running conditions were mobile phase A (water + acetic acid 0.1%), mobile phase B (methanol 100%), mobile phase C (acetonitrile 100%) with a 0.3 mL/min flow and a total running time of 14 min with temperatures 35 and 20 °C for the column and the sample, respectively, The absorbance was read at 280 nm. An elution gradient was used starting with 90% (A), 5% (B) 5% (C); changing the proportion to the minute 6 with 76% (A), 12% (B) y 12% (C); to the minute 11 changed it with 36% (A), 32% (B) and 32% (C), to change the initial gradient 90% (A), 5% (B) 5% (C) to 12 min until finish. Data based on polyphenol content in foods [53] was used to identify phenolic compounds and verify them by mass spectrometry from 100 to 750 m/z in negative ionization mode with a cone voltage of 30 kV and a desolvation temperature of 400 °C.

### 3.11. Statistical Analysis

Multifactorial analysis was performed with three replicates. The InfoStat Version 2008 software package was used for analysis of variance (ANOVA). Comparison of means was performed by the Tukey’s test. A level of *p* < 0.005 was set as of statistical significance.

## 4. Conclusions

Raw Kabuli-type showed excellent quality and high phenolic content and antioxidant capacity. The cooked chickpea showed high nutritional and technological quality, in addition to maintaining its phenolic composition and antioxidant capacity during storage under a nitrogen atmosphere at low temperatures (4 and −20 °C) for 50 days. These results may have a commercial impact and may have greater economic benefit by increasing the storage of cooked chickpea using controlled atmosphere technology considered economical and friendly to the environment. On the other hand, free and conjugated phenolic compounds showed more bioaccessibility than bound phenolics observed in the in vitro digestion. Therefore, chickpea contains compounds of biological interest with antioxidant potential, suggesting that this legume can be exploited for various technologies. However, it will be necessary in subsequent studies to focus more on the release of the bound phenolic compounds since they are found in greater quantity and confer the greatest antioxidant capacity to take full advantage of the properties of these compounds from chickpea.

## Figures and Tables

**Figure 1 molecules-26-02773-f001:**
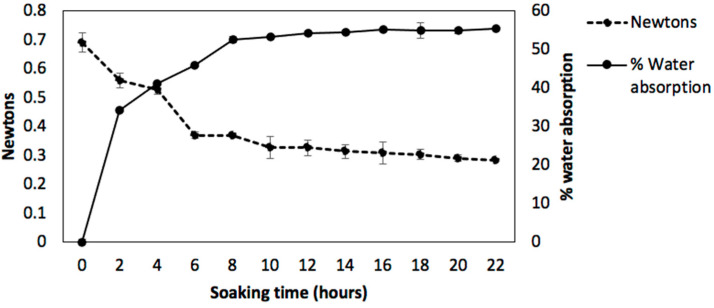
Water absorption and texture during the soaking time of raw chickpea.

**Figure 2 molecules-26-02773-f002:**
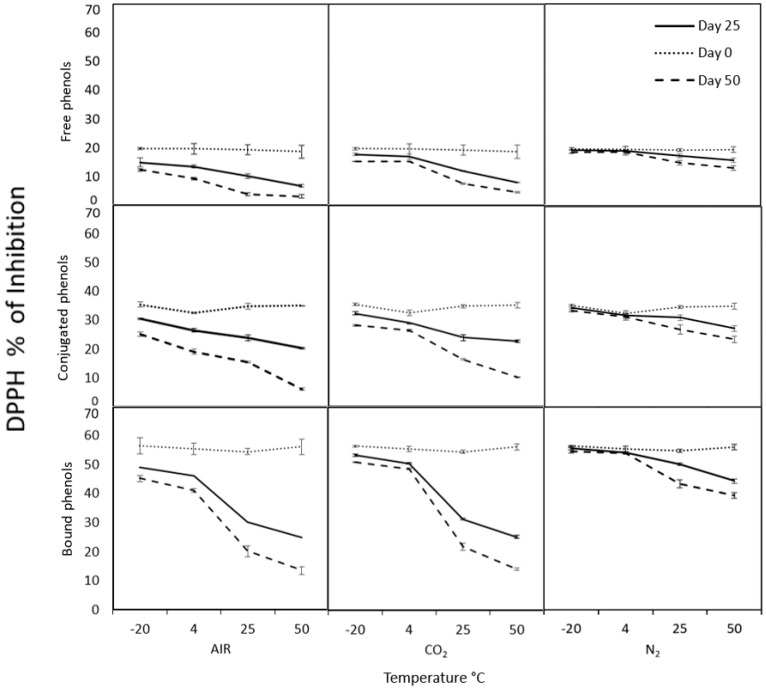
Antioxidant capacity by radical DPPH of free, conjugated and bound phenols under different storage conditions. Control was air treatment.

**Figure 3 molecules-26-02773-f003:**
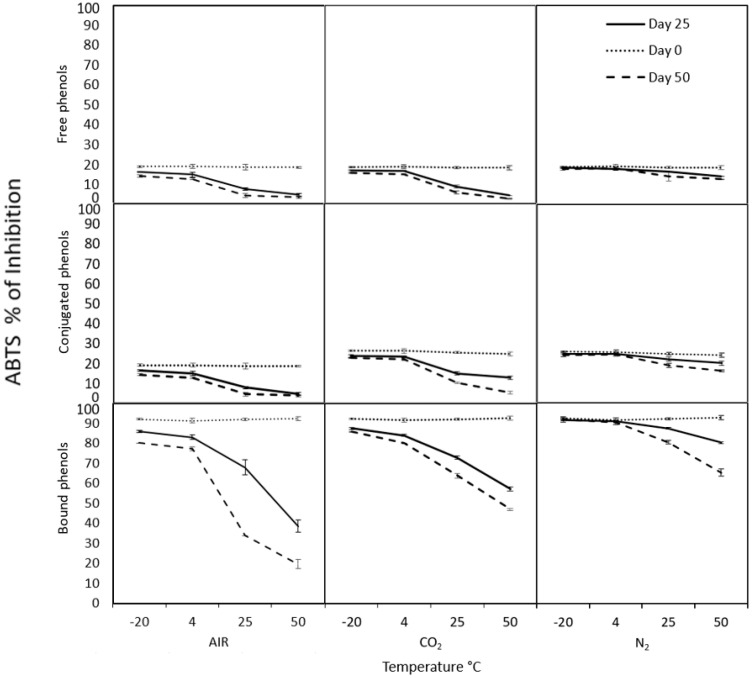
Antioxidant capacity by radical ABTS of free, conjugated, and bound phenols under different storage conditions. Control was air treatment.

**Figure 4 molecules-26-02773-f004:**
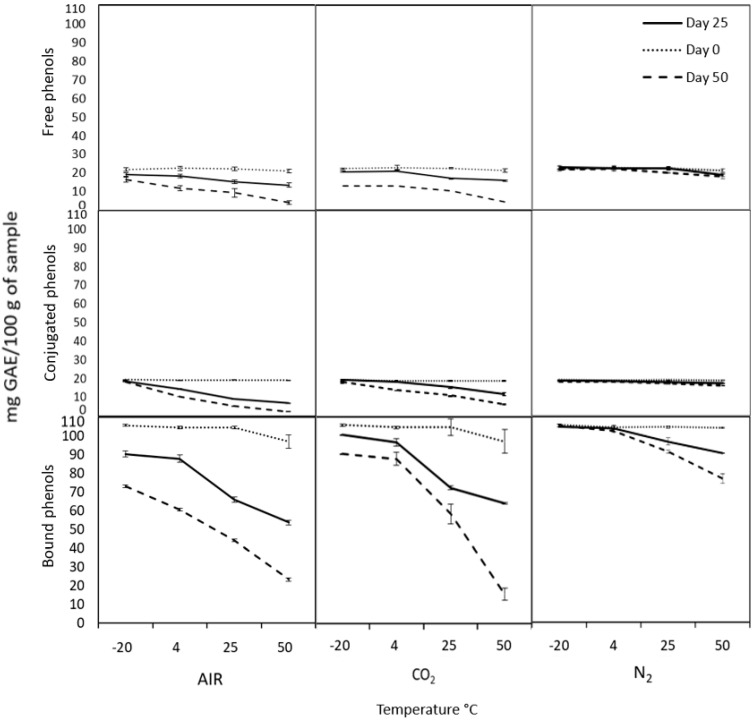
Quantification of total phenols of free, conjugated and bound phenols under different storage conditions. Control was air treatment.

**Figure 5 molecules-26-02773-f005:**
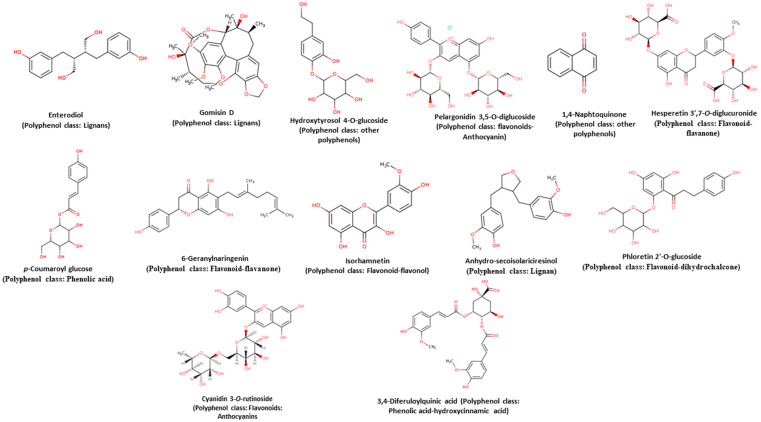
Structures of phenolic compounds from chickpea identified by UPLC-MS. The images were taken from the database on polyphenol content in foods [53].

**Figure 6 molecules-26-02773-f006:**
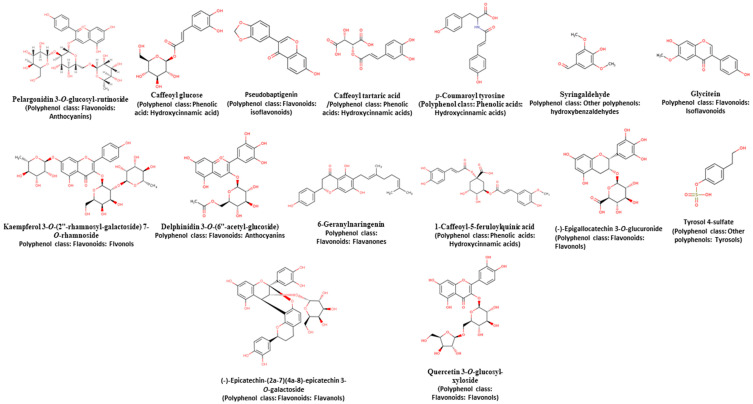
Structures of phenolic compounds from chickpea identified by UPLC-MS in the in vitro gastrointestinal digestion. The images were taken from the database on polyphenol content in foods [53].

**Table 1 molecules-26-02773-t001:** Proximal composition of raw and cooked chickpea.

COMPOSITION (%)	RAW	COOKED
Carbohydrate	64.40 ^a^ ± 0.5	68.32 ^a^ ± 0.34
* Moisture	5.96 ^a^ ± 0.10	47.28 ^b^ ± 0.07
Ash	2.8 ^a^ ± 0.47	1.75 ^b^ ± 0.04
Protein	19.16 ^a^ ± 0.51	18.38 ^b^ ± 1.98
Fat	6.23 ^a^ ± 0.07	5.35 ^b^ ± 0.08
Fiber	7.41 ^a^ ± 0.97	7.2 ^a^ ± 0.82

***** Wet sample; Mean values of standard deviation based on triplicate determination; Means in the same line with different letters are significantly different; (*p* < 0.05).

**Table 2 molecules-26-02773-t002:** Color in raw and cooked chickpea.

Sample	L*	a*	b*
Raw	60.44 ± 3.5 ^a^	6.98 ± 0.4 ^a^	17.74 ± 1.25 ^a^
Cooked	55.45 ± 3.81 ^b^	8.63 ± 0.62 ^b^	22.4 ± 1.9 ^b^

Mean standard deviation based in 17 determinations. Different letters in each column indicate significant difference (*p* < 0.05), Tukey range test. L*: luminosity (100: white to 0: black); a* indicate red (+a*) and (−a*) green shades; b* indicate yellow (+b*) and blue (−b*) shades.

**Table 3 molecules-26-02773-t003:** Proximal chemical analysis in cooked chickpea at −20 °C in N_2_ atmosphere at
day 0 and day 50.

COMPOSITION (%)	DAY 0	DAY 50
Carbohydrate	68.32 ^a^ ± 0.34	67.29 ^a^ ± 0.65
Ash	1.75 ^a^ ± 0.04	2.05 ^a^ ± 0.03
Protein	18.38 ^a^ ± 1.98	17.18 ^a^ ± 2.06
Fat	5.35 ^a^ ± 0.08	5.45 ^a^ ± 0.02
Fiber	7.20 ^a^ ± 0.82	7.30 ^a^ ± 0.60

Mean values of standard deviation based on triplicate determination. Means in the same line with different letters are significantly different (*p* < 0.05).

**Table 4 molecules-26-02773-t004:** Antioxidant capacity and phenol compounds quantification during in vitro digestion process.

Antioxidant Technique/Phenolic Fraction	Raw	Cooked	Mouth *	Stomach *	Small Intestine *	
			(Alpha-Amylase)	(Pepsin)	(Pancreatin) Inside Membrane	(Portal Vein) Outside Nembrane
**DPPH ****						
Free	16.8 ^aA^ ± 1.5	8.2 ^bA^ ± 0.5	0.6 ^cA^ ± 0.1	3.7 ^deA^ ± 1.2	4.9 ^eA^ ± 0.4	3.0 ^dA^ ± 0.4
Conjugated	17.3 ^aA^ ± 0.7	9.7 ^bA^ ± 0.3	0.17 ^cB^ ± 0.0	2.5 ^dA^ ± 0.0	6.7 ^eA^ ± 1.0	5.0 ^eB^ ± 1.0
Bound	35.1 ^aB^ ± 3.7	26.8 ^bB^ ± 1.2	0.0 ^cB^ ± 0.0	9.8 ^dB^ ± 1.7	16.2 ^eB^ ± 0.5	0.0 ^cC^ ± 0.0
**ABTS ****						
Free	15.6 ^aA^ ± 1.4	8.7 ^bA^ ± 0.2	1.9 ^cA^ ± 0.6	3.2 ^cA^ ± 1.1	8.3 ^cA^ ± 0.1	6.7 ^bA^ ± 0.1
Conjugated	19.6 ^aA^ ± 4.3	9.3 ^bA^ ± 0.0	2.3 ^cB^ ± 0.1	6.9 ^bcA^ ± 2.4	8.3 ^bcA^ ± 0.9	5.3 ^bcAB^ ± 1.2
Bound	41.8 ^aB^ ± 2.1	21.7 ^bB^ ± 0.2	0.75 ^cC^ ± 0.0	8.1 ^dA^ ± 2.1	11.1 ^dB^ ± 0.1	2.6 ^cB^ ± 0.1
**FRAP ****						
Free	18.7 ^aA^ ± 0.2	7.2 ^bA^ ± 0.1	4.5 ^cA^ ± 0.2	3.3 ^dA^ ± 0.4	6.3 ^eA^ ± 0.1	5.1 ^cA^ ± 0.3
Conjugated	66.2 ^aB^ ± 0.2	41.0 ^bB^ ± 3.1	3.4 ^cB^ ± 0.3	11.7 ^dB^ ± 0.1	32.0 ^eB^ ± 0.2	28.2 ^fB^ ± 0.5
Bound	71.2 ^aC^ ± 0.1	30.2 ^bC^ ± 0.2	3.6 ^cB^ ± 0.1	6.8 ^dC^ ± 1.1	13.5 ^eC^ ± 0.1	3.9 ^cC^ ± 0.1
**AAPH ****						
Free	11.3 ^aA^ ± 1.3	6.1^bA^ ± 1.7	0.5 ^cA^ ± 0.0	2.6 ^cdA^ ± 0.7	5.1 ^dbA^ ± 0.7	4.6 ^dbA^ ± 1.5
Conjugated	67.7 ^aB^ ± 4.0	31.9 ^bB^ ± 1.6	3.8 ^cB^ ± 0.4	11.0 ^dB^ ± 0.4	30.6 ^bB^ ± 0.7	19.0 ^eB^ ± 0.4
Bound	76.8 ^aC^ ± 1.6	43.0 ^bC^ ± 0.7	2.5 ^cC^ ± 0.4	7.4 ^dC^ ± 0.7	9.7 ^eA^ ± 0.7	0.0 ^fC^ ± 0.0
**Phenol Compounds *****						
Free	57.5 ^aA^ ± 5.0	23.7 ^bA^ ± 1.0	3.0 ^eA^ ± 0.1	5.5 ^dA^ ± 0.3	7.7 ^cA^ ± 0.6	7.1 ^cA^ ± 0.1
Conjugated	27.2 ^aB^ ± 0.1	19.4 ^bA^ ± 0.0	1.8 ^cB^ ± 0.1	5.5 ^dA^ ± 0.3	9.0 ^eA^ ± 1.1	8.8 ^eA^ ± 0.1
Bound	146.0 ^aC^ ± 4.0	106.6 ^bB^ ± 4.0	4.2 ^cC^ ± 0.5	13.9 ^dB^ ± 2.6	39.3 ^eB^ ± 0.6	1.4 ^fB^ ± 0.1

Media ± Standard deviation based on three replicas. Different lowercase letters in row indicate that there was a significant difference (*p* < 0.05). Different uppercase letters in column indicate that there was a significant difference (*p* < 0.05) in each antioxidant capacity assay. Controls in DPPH, ABTS, FRAP and phenol compounds determination were the reagents with the solvent of the samples. Controls in AAPH were a suspension of erythrocytes, and AAPH with erythrocytes. * measurements were made on cooked chickpea. ** Reported in nmol TE/g of sample. *** Reported in mg GAE/100 g of sample.

**Table 5 molecules-26-02773-t005:** Free, conjugated and bound phenolic compounds identified by UPLC-MS from raw and cooked chickpea.

Sample/Phenolic Compounds	No.	*RT* (min)	% of Area	Compound *	Observed *m/z*	Exp.^a^ Mass (Da)	Theor.^b^ Mass (Da)	Mass Delta (Da)	Molecular Formula
Raw chickpea/Free	1	0.63	6.40	Gomisin D	529.33	530.33	530.21	0.12	C_28_H_34_O_10_
	2	0.99	46.40	Enterodiol	301.33	302.33	302.15	0.18	C_18_H_22_O_4_
	3	9.71	3.08	Hydroxytyrosol 4-*O*-glucoside	315.18	316.18	316.12	0.06	C_14_H_20_O_8_
	4	9.91	8.67	Pelargonidin 3,5-*O*-diglucoside	629.35	630.35	630.13	0.22	C_27_H_31_ClO_15_
	5	10.06	3.54	1,4-Naphtoquinone	157.18	158.18	158.04	0.14	C_10_H_6_O_2_
	6	11.43	2.90	*p*-Coumaroyl glucose	325.19	326.19	326.10	0.09	C_15_H_18_O_8_
Raw chickpea/Conjugated	1	0.55	7.46	6-Geranylnaringenin	407.27	408.27	408.19	0.08	C_25_H_28_O_5_
	2	0.63	9.72	Gomisin D	529.33	530.33	530.21	0.12	C_28_H_34_O_10_
	3	0.99	57.12	Enterodiol	301.33	302.33	302.15	0.18	C_18_H_22_O_4_
	4	9.71	1.55	Isorhamnetin	315.11	316.11	316.05	0.06	C_16_H_12_O_7_
	5	9.97	1.95	Pelargonidin 3,5-*O*-diglucoside	629.48	630.48	630.13	0.35	C_27_H_31_ClO_15_
Raw chickpea/Bound	1	0.56	4.50	Hydroxytyrosol 4-*O*-glucoside	315.31	316.31	316.11	0.20	C_14_H_20_O_8_
	2	0.64	6.75	Gomisin D	529.33	530.33	530.21	0.12	C_28_H_34_O_10_
	3	0.99	55.29	Enterodiol	301.33	302.33	302.15	0.18	C_18_H_22_O_4_
	4	1.57	12.43	Anhydro-secoisolariciresinol	343.38	344.38	344.16	0.22	C_20_H_24_O_5_
	5	9.72	1.67	Phloretin 2′-*O*-glucoside	434.96	435.96	436.13	−0.17	C_21_H_24_O_10_
Cooked chickpea/Free	1	0.55	5.62	Hydroxytyrosol 4-*O*-glucoside	315.31	316.31	316.11	0.20	C_14_H_20_O_8_
	2	0.63	10.33	Gomisin D	529.33	530.33	530.21	0.12	C_28_H_34_O_10_
	3	0.99	59.15	Enterodiol	301.20	302.20	302.15	0.05	C_18_H_22_O_4_
	4	3.39	5.00	Hesperetin 3′,7-*O*-diglucuronide	653.27	654.27	654.14	0.13	C_28_H_30_O_18_
	5	9.72	4.42	Hydroxytyrosol 4-*O*-glucoside	315.05	316.05	316.12	−0.07	C_14_H_20_O_8_
	6	9.94	2.38	Pelargonidin 3,5-*O*-diglucoside	629.55	630.55	630.13	0.42	C_27_H_31_ClO_15_
Cooked chickpea/Conjugated	1	0.55	7.73	6-Geranylnarin-genin	407.40	408.40	408.19	0.21	C_25_H_28_O_5_
	2	0.63	10.09	Gomisin D	529.46	530.46	530.21	0.25	C_28_H_34_O_10_
	3	0.99	67.42	Enterodiol	301.20	302.20	302.15	0.05	C_18_H_22_O_4_
Cooked chickpea/Bound	1	0.52	3.87	Hydroxytyrosol 4-*O*-glucoside	315.37	316.37	316.11	0.26	C_14_H_20_O_8_
	2	0.62	5.81	Gomisin D	529.39	530.39	530.21	0.18	C_28_H_34_O_10_
	3	0.99	55.62	Enterodiol	301.40	302.40	302.15	0.25	C_18_H_22_O_4_
	4	1.56	12.78	Anhydro-secoisolariciresinol	343.38	344.38	344.16	0.22	C_20_H_24_O_5_
	5	9.71	6.61	3,4-Diferuloylquinic acid	543.43	544.43	544.16	0.27	C_27_H_28_O_12_

**^a^** Experimental; **^b^** Theoretical; * Identification based on data base on polyphenol content in foods [53].

**Table 6 molecules-26-02773-t006:** Phenolic compounds identified by UPLC-MS during in vitro digestion.

Simulated Gastrointestinal Tract Section/Phenolic Compounds	No.	RT (min)	% of Area	Compound *	Observed *m/z*	Exp.^a^ Mass (Da)	Theor.^b^ Mass (Da)	Mass Delta (Da)	Molecular Formula
Mouth/Free	1	9.92	7.93	Caffeoyl tartaric acid	311.08	312.08	312.04	0.04	C_13_H_12_O_9_
	2	10.36	1.87	*p*-Coumaroyl tyrosine	326.23	327.23	327.11	0.12	C_18_H_17_NO_5_
Mouth/Conjugated	Not identified								
Mouth/Bound	1	11.45	13.27	Syringaldehyde	181.15	182.15	182.06	0.09	C_9_H_10_O_4_
	2	11.79	9.82	Glycitein	283.40	284.40	284.07	0.33	C_16_H_12_O_5_
Stomach/Free	1	8.18	12.36	Kaempferol 3-*O*-(2″-rhamnosyl-galactoside) 7-*O*-rhamnoside	738.93	739.93	740.21	−0.28	C_33_H_40_O_19_
	2	10.00	25.43	Delphinidin 3-*O*-(6″-acetyl-glucoside)	506.52	507.52	507.11	0.41	C_23_H_23_O_13_
Stomach/Conjugated	1	1.8	1.84	6-Geranylnaringenin	407.40	408.40	408.19	0.21	C_25_H_28_O_5_
	2	9.91	6.89	1-Caffeoyl-5-feruloylquinic acid	529.46	530.46	530.14	0.32	C_26_H_26_O_12_
Stomach/Bound	1	9.43	5.44	(-)-Epigallocatechin 3-*O*-glucuronide	481.10	482.10	482.10	0.00	C_21_H_22_O_13_
	2	9.99	5.39	Delphinidin 3-*O*-(6″-acetyl-glucoside)	506.52	507.52	507.11	0.41	C_23_H_23_O_13_
	3	10.3	9.54	Cyanidin 3-*O*-rutinoside	594.26	595.26	595.16	0.10	C_27_H_31_O_15_
	4	10.47	3.76	Quercetin 3-*O*-glucosyl-xyloside	594.65	595.65	596.13	−0.48	C_26_H_28_O_16_
Intestine/Free	1	10.3	3.93	Tyrosol 4-sulfate	216.97	217.97	218.02	−0.05	C_8_H_10_O_5_S
	2	10.5	2.67	(-)-Epicatechin-(2a-7)(4a-8)-epicatechin 3-*O*-galactoside	705.13 (18)	706.13	706.19	−0.06	C_36_H_34_O_15_
Intestine/Conjugated	1	8.2	10.90	Pelargonidin 3-*O*-glucosyl-rutinoside	739.91	740.91	741.22	−0.31	C_33_H_41_O_19_
Intestine/Bound	1	1.85	6.82	Caffeoyl glucose	341.43	342.43	342.09	0.34	C_15_H_18_O_9_
	2	4.06	1.98	Pseudobaptigenin	281.38	282.38	282.05	0.33	C_16_H_10_O_5_
	3	9.98	8.40	Delphinidin 3-*O*-(6″-acetyl-glucoside)	506.39	507.39	507.11	0.28	C_23_H_23_O_13_
	4	10.3	6.25	Tyrosol 4-sulfate	217.04	218.04	218.02	0.02	C_8_H_10_O_5_S

**^a^** Experimental; **^b^** Theoretical; * Identification based on data base on polyphenol content in foods [53].

## Data Availability

The data generated from this research are available from the authors.

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
