# Peer review of "Evaluation of Quality, Antioxidant Capacity, and Digestibility of Chickpea (Cicer arietinum L. cv Blanoro) Stored under N2 and CO2 Atmospheres"

_molecules, 2021, doi:10.3390/molecules26092773_

Round 1
Reviewer 1 Report
Dear Authors,
After the review process, I have several comments: you should clearly present the aim of the paper in the abstract; you should change the phrase - lie31-34, it is not clear what they want to say; you should include in the introduction new data about in vitro researches; for example, new method about bioavailability and bioactivities of polyphenols after in vitro digestion or how to prove the bioactive potential of functional products and bioavailability of phenolic compounds; you should check Line 99; you should define the controls in Materials and Methods and include it at the Tables/Figures legends; you should include the chromatograms for the main compounds as a supplementary data.
Best regards,
Author Response
Author: We thank the reviewer for the comments.
Reviewer 1: After the review process, I have several comments: you should clearly present the aim of the paper in the abstract you should change the phrase - lie31-34, it is not clear what they want to say;
Author: The objective in the abstract was added lines 25-26(The aim of this work was to monitor the quality, antioxidant capacity and digestibility of chickpea exposed to different modified atmospheres.) We changed the phrase, lines 32-35(After in vitro digestion, the antioxidant capacity was measured by DPPH, ABTS, FRAP (ferric reducing antioxidant power), and AAPH (2,2’-Azobis [2-methylpropionamidine]). Additionally, quantification of total phenols, and UPLC-MS profile were determined.)
Reviewer 1: you should include in the introduction new data about in vitro researches; for example, new method about bioavailability and bioactivities of polyphenols after in vitro digestion or how to prove the bioactive potential of functional products and bioavailability of phenolic compounds you should check Line 99.
Author: We included new data, lines 94-114:
The multiple advantages of in vitro research, such as being fast and cheap, better condition control, and less ethical restrictions, have made it an alternative method to studying the metabolic process of biologically active substances in vivo [19]. Recently studies have focused on determining the bioaccessibility and bioavailability of phenolic compounds in legumes through static or dynamic in vitro models. The static model preferably is used if the focus of research is on hypothesis building and a large number of samples is to be analyzed, while the dynamic model is used if closely simulating physiological conditions is the primary aim [20]. For polyphenols, there is limited evidence as to which method is the most appropriate for measuring bioaccessibility. However, to test the bioactivity of these antioxidant compounds after in vitro digestion some antioxidant techniques such as DPPH (2,2’-diphenyl-1-picrylhydrazyl), ABTS (2,2’-azino-bis-[3ethylbenzothiazoline-6-sulfonic acid]) and FRAP (ferric reducing antioxidant power) have been used [19 , 21, 22]. The determination of phenolic compounds in each step of digestion is commonly used to measure their bioaccessibility [23, 24]. Bioavailability has been studied by quantifying these compounds after being absorbed by a cellulose membrane used to simulate the small intestine [25, 26] or also using Caco-2 cell monolayers [21,27].
Reviewer 1: you should define the controls in Materials and Methods and include it at the Tables/Figures legends
Author: In the controlled atmospheres, the control was the sample that contained air, this was already indicated in the methodology (lines 540-541) and in the footer of Figures 2, 3 and 4 “control was air treatment”. Controls in DPPH, ABTS, FRAP and phenol compounds determination were the reagents with the solvent of the samples. Controls in AAPH were a suspension of erythrocytes, and AAPH with erythrocytes. These were included in methodology (lines 601, 613, 621638-639, 649) and table 4. The rest of the methodologies compared raw with cooked chickpea.
Reviewer 1: you should include the chromatograms for the main compounds as a supplementary data
Author: The chromatograms were included as a supplementary data available on line at https://unisonmx-my.sharepoint.com/:f:/g/personal/carmen_deltoro_unison_mx/Elqb_1p2mblGlmsWFxhW3NgB_VxIuS5P209UIU-jpXUHCw?e=FZjbw3, Figure S1: UPLC-MS chromatograms of the phenolic compounds identified in raw and cooked chickpea. Figure S2: UPLC-MS chromatograms of the phenolic compounds identified in the mouth. Figure S3: UPLC-MS chromatograms of the phenolic compounds identified in the stomach. Figure S4: UPLC-MS chromatograms of the phenolic compounds identified in the intestine.
This was also indicated within the text, lines 432 (Figures S2-S4).
Additional modifications
Author: Changes or new information added to the document can be seen in yellow. Reorganization in the numbering of the references and numbering of figures and tables can be seen in blue.
Reviewer 2 Report
The study of Perez-Perez et al. aimed to determine the quality and antioxidant capacity of free, conjugated and bound cooked chickpea Kabuli-type under atmospheres of N2 and CO2 and its digestibility through a gastrointestinal system in vitro. This paper is clear, well prepared, and organized with high scientific quality and relevance. The presented topic is in line with the current trends in food technology.
The Folin-Cocialteu method ensures low specificity and the possibility of reacting the reagent not only with polyphenols, but also with other compounds (e.g. proteins, vitamins), so the results obtained may be encumbered with errors. Please mention this in the discussion.
Table 5 provides information that could be included in supplementary materials.
Author Response
Reviewer 2: The study of Perez-Perez et al. aimed to determine the quality and antioxidant capacity of free, conjugated and bound cooked chickpea Kabuli-type under atmospheres of N2 and CO2 and its digestibility through a gastrointestinal system in vitro. This paper is clear, well prepared, and organized with high scientific quality and relevance. The presented topic is in line with the current trends in food technology.
Author: We thank the reviewer for the comments.
Reviewer 2: The Folin-Cocialteu method ensures low specificity and the possibility of reacting the reagent not only with polyphenols, but also with other compounds (e.g. proteins, vitamins), so the results obtained may be encumbered with errors. Please mention this in the discussion.
Author: We included this discussion, lines 318-322 (In this context, the Folin-Ciocalteu technique used for the determination of polyphenols can react with other compounds different from phenolics such as proteins, sugars and others with reducing power, which may be present in chickpea extracts [50]).
Reviewer 2: Table 5 provides information that could be included in supplementary materials.
Author: This table was included as a supplementary data available on line at https://unisonmx-my.sharepoint.com/:f:/g/personal/carmen_deltoro_unison_mx/Elqb_1p2mblGlmsWFxhW3NgB_VxIuS5P209UIU-jpXUHCw?e=FZjbw3, Table S1: Pearson´s correlation coefficients of Total phenolic compounds versus raw and cooked chickpea, with the different antioxidant techniques (ABTS, DPPH, FRAP, AAPH).
Additional modifications
Author: Changes or new information added to the document can be seen in yellow. Reorganization in the numbering of the references and numbering of figures and tables can be seen in blue.
Round 2
Reviewer 1 Report
No other comments.